# Martensitic Phase-Transforming Metamaterial: Concept and Model

**DOI:** 10.3390/ma16216854

**Published:** 2023-10-25

**Authors:** Sosuke Kanegae, Masayuki Okugawa, Yuichiro Koizumi

**Affiliations:** Division of Materials and Manufacturing Science, Graduate School of Engineering, Osaka University, 2-1 Yamadaoka, Suita 565-0871, Osaka, Japan; sosuke.kanegae@mat.eng.osaka-u.ac.jp

**Keywords:** metamaterial, martensitic transformation, superelasticity, shape memory effect

## Abstract

We successfully developed a mechanical metamaterial that displays martensitic transformation for the first time. This metamaterial has a bistable structure capable of transitioning between two stable configurations through shear deformation. The outer shape of the unit cell of this structure is a parallelogram, with its upper and lower sides forming the bases of two solid triangles. The vertices from these triangles within the parallelogram are linked by short beams, while the remaining vertices are linked by long beams. The elastic energy of the essential model of the metamaterial was formulated analytically. The energy barrier between these two stable configurations consists of the elastic strain energy due to the tensile deformation of the short beams, the compressive deformation of the long beams, and the bending deformation of the connecting hinges. One example of a novel metamaterial was additively manufactured via the materials extrusion (MEX) process of thermoplastic polyurethane. The metamaterial exhibited deformation behaviors characteristic of martensitic transformations. This mechanical metamaterial has the potential to obtain properties caused by martensitic transformation in actual materials, such as the shape memory effect and superelasticity.

## 1. Introduction

Crystalline materials consist of atoms arranged spontaneously in patterns with a crystal symmetry. Depending on the combination of atom types and crystallographic symmetry, the material exhibits various properties, such as hardness, color, thermal conductivity, and shape memory effect. Metamaterials [1] are, in general, artificial materials that exhibit material properties that the spontaneous atomic arrangement in conventional materials processing cannot achieve. Metamaterials are categorized based on the specific properties they are engineered to manipulate. These categories include mechanical metamaterials [2], thermal metamaterials [3,4], acoustic metamaterials [5], and more.

Generally, metamaterials are designed by the combination of the design of a unit structure, so-called “meta-atoms” [6], and the periodic arrangement of the meta-atoms with crystallographic symmetry in the same manner as actual atoms in crystalline materials. We propose a new design for metamaterials mimicking actual materials that exhibit unique properties, intending to develop a technology that can contribute to a reduction in the use of rare elements. Our concept is as follows: The properties of actual materials originate from the combination of atomic-scale and subatomic-scale structures, such as bonding, and the periodic arrangement of the unit structure with crystallographic symmetry.

Metamaterials, in principle, attract attention for their novel properties that cannot be achieved by existing materials [7]. Again, we would like to emphasize that it will be possible to reproduce the unique properties that have previously only been achieved by using materials containing rare elements, by using ubiquitous elements, and by designing metamaterials by carefully analyzing the relationship between the basic structures of atoms and their properties.

Phase transformation is an important factor for materials to develop some functional material properties. Martensitic transformation, in particular, is closely related to functional properties such as the quench hardening of steels [8,9,10], shape memory effect [11,12,13], and superelasticity [12,14,15]. Martensitic transformation is a shear displacive phase transformation without diffusion. It is expected that the shape memory effect can be reproduced in structures by designing metamaterials focused on the atomic arrangement change due to shear, which is the elementary process of martensitic transformation.

Metamaterials that reproduce phase transitions in meta-atoms were published in 2015 by Restrepo et al. [16]. The materials were named “Phase Transforming Cellular Materials (PXCM). Conventional phase-transforming metamaterials (i.e., PXCM) deform in one direction in an expanding and contracting manner in response to tensile and compressive applied loads [17,18,19,20,21,22]. Some phase-transforming metamaterials exhibit bistability in deformation in a unique direction [23,24]. The transformation in these structures is a type of diffusionless phase transition. In our previous study [25], we developed a novel PXCM that can exhibit the expanding-and-contracting-type phase transformation in four crystallographically equivalent directions in response to a tensile and compressive applied load. The PXCM was designed by utilizing the crystallographic symmetry of a face-centered cubic (FCC) structure and choosing the four <1 1 1> directions as the loading axes. This was the first demonstration of the concept of “Atom-Mimetics”, in which the structure and behaviors of atoms are used as a guide for the development of a novel metamaterial. The new PXCM was named multi-axis PXCM. Also, we developed a type of PXCM that can transform in response to changes in temperature by utilizing a bimetal beam that bended spontaneously with an increasing temperature [26]. The PXCM was named Thermally Induced PXCM (TI-PXCM). The transforming behaviors were qualitatively similar to those achieved by the diffusive transformation of actual crystalline materials. Nevertheless, their quantitative properties are quite different from existing materials. The properties, such as stress-induced transformation and transformation-induced thermal expansion, can be far beyond the limit of the actual materials. It was demonstrated that negative large thermal expansion is possible.

However, the transformations achieved with the previously developed PXCMs are not martensitic transformations. Although many such phase-transforming metamaterials and shear-deforming metamaterials [27,28] have been developed, there are no examples of metamaterials that reproduce phase transitions via shear deformation. A novel PXCM that exhibits martensitic transformation has great potential for the development of more functional PXCMs.

This study explores the design of metamaterials, focusing on the change in the atomic arrangement due to shear. As a basic guideline for the design of metamaterials, we investigated the constraints on the design of two-dimensional lattice structures with bistable structures, in which metastable states appear owing to shear deformation.

## 2. Modeling Approach

Alqasimi [29] presented a bistable element of bar-constrained quadrilateral linkage. In the present study, we introduce quadrilateral linkage as a parallelogram ABDC, as shown in Figure 1, to be periodically arranged as a two-dimensional metamaterial. *L* represents the length of the AC side of the parallelogram, and *l* represents the length of the P’Q side. Let triangle P’CD with the CD side of the parallelogram ABDC be congruent to the triangle PAB with the AB side of the parallelogram ABDC. In this case, when each node of the linkage of the parallelogram ABDC rotates freely, point P’ moves freely on a circle with the center at point P and a radius L. Also, assume that triangle PBA with the BA side as one of the sides of the parallelogram ABDC is congruent to the triangle QAB. If point Q and point P’ are bounded by a bar of length l, point P’ moves freely on a circle with a radius l centered at point Q when the nodes are free to rotate. This parallelogram linkage is a stable structure when point P’ exists on two circles, i.e., at point P’ or point P”, the intersection of the two circles. In other words, this parallelogram linkage is a bistable structure that is stabilized in two states: the parallelogram ABDC state and the parallelogram ABD’C’ state.

When manufacturing this structure as a metamaterial, the structure must be seamless. Therefore, the free-rotation nodes assumed above cannot be incorporated into the design. In this study, instead of free-rotation nodes, we designed the structure as a flexure hinge [30,31,32], which is a rectangular beam with circular arcs on both sides to make it bend easily, as shown in Figure 2. In this study, the elastic properties of the flexure hinge were obtained using the approximate formula (Equation (1)) of Schotborgh et al. [33]. In this approximate formula, the hardness is determined from the radius of the arc *R*, the thickness of the neck of the flexure hinge *t*_hinge_, the inclination angle around the normal direction (ND) *α*_ND_, the inclination angle around the ND *M*_ND_, the Young’s modulus *E*, and the depth of the beam *b* (the use of these symbols aligns with Ref. [11]).
(1)$$\frac{\alpha_{ND}}{M_{ND}}={\left\{\frac{Ebt_{hinge}^{2}}{12}\left[-0.0089+1.3556\sqrt{\frac{t_{hinge}}{2R}}-0.5227{\left(\sqrt{\frac{t_{hinge}}{2R}}\right)}^{2}\right]\right\}}^{-1}$$

We designed a two-dimensional periodic structure using the bistable structure of Alqasimi [29] and the flexure hinge of Schotborgh et al. [33], as shown in Figure 3. The red line in the figure shows the bistable structure of Alqasimi et al. [29] and the flexure hinge of Schotborgh et al. [33]. The red dashed line connects the midpoint of the edge AB (CD) and the vertex Q (P’). The length of these dashed line segments is defined as *h*. 

The parallelogram edges AC and BD are defined as “long beams” and the bar P’Q is defined as a “short beam”.

## 3. Method

We examined each dimensional change in this structure when it was sheared from its initial shape. First, in Figure 4a, the length variation ∆*l* in the short beam and the angle variation ∆*φ* around point Q are obtained when the length *L* of the long beam is constant, and the AC side rotates ∆*θ* around point A.

The deforming parts can be extracted, as shown in Figure 4b, assuming that the parts of triangle P’AB and triangle QBA are rigid. In order to understand the change in the short beam relative to the change in the long beam easily, first, the long beam is assumed to be rigid. The length change in the short beam ∆*l* and the inclination angle ∆*φ* of the short beam are described as a function of the inclination angle ∆*θ* of the long beam for the case where the angle of the parallelogram changes from α to β. The end-point z of the long beam rotates around the opposite end-point O. Meanwhile, the line segment O’-z represents the angle and the length of the short beam. The point w is obtained by rotating point z by Δ*θ* around point O. Figure 4b can be used to derive the relationships among the points, the inclination angles of the beams, and the lengths of the beams during the deformation process. The positions of point O, O’, z, and w are described on a complex plane, taking point O as the origin and the direction from O to z as the positive direction of the real axis. Then, each point is represented by Equation (2).
(2a)$$O=\left(0,0\right)=0+0\;i=0$$
(2b)$$O^{\prime }=\left(2hcos\left(\frac{\beta -\alpha }{2}\right),2hsin\left(\frac{\beta -\alpha }{2}\right)\right)=2h\left\{cos\left(\frac{\beta -\alpha }{2}\right)+isin\left(\frac{\beta -\alpha }{2}\right)\right\}=2he^{i\frac{\beta -\alpha }{2}}$$
(2c)$$z=\left(L,0\right)=L+0\;i=L$$
(2d)$$w=\left(Lcos\Delta \theta ,Lsin\Delta \theta \right)=Lcos\Delta \theta +iLsin\Delta \theta =Le^{i\Delta \theta }$$

In Equation (2), the positions of the points are expressed with the conventional Cartesian coordinate, followed by the expression with the complex plane, as well for convenience in understanding,

The initial length of the short beam is expressed by Equation (3).
(3)$$l=\left|z-O^{\prime }\right|=\sqrt{L^{2}+4h^{2}-2hL\cos\frac{\beta -\alpha }{2}}$$

The length of the short beam due to the declination angle of the long beam ∆*θ* is expressed by Equation (4).
(4)$$l^{\prime }=\left|w-O^{\prime }\right|=\sqrt{L^{2}+4h^{2}-2hL\cos\left(∆\theta -\frac{\beta -\alpha }{2}\right)}$$

Therefore, the length change in the short beam ∆*l* is expressed by Equation (5).
(5)$$\begin{array}{l} ∆l=l^{\prime }-l\\ \;\;\;\;=\sqrt{L^{2}+4h^{2}-2hL\cos\left(∆\theta -\frac{\beta -\alpha }{2}\right)}-\sqrt{L^{2}+4h^{2}-2hL\cos\frac{\beta -\alpha }{2}} \end{array}$$

In order to describe the length of the short beam as a function of the inclination angle, Δ*φ*, the trajectory of point w with respect to point O’, expressed by Equation (6), can be used.
(6)$$\frac{l^{\prime }}{l}e^{i\Delta \varphi }=\frac{w-O^{\prime }}{z-O^{\prime }}=\frac{Le^{i\Delta \theta }-2he^{i\frac{\beta -\alpha }{2}}}{L-2he^{i\frac{\beta -\alpha }{2}}}$$

Then, the ∆φ is expressed by Equation (7),
(7)$$∆\varphi =\arg\left(\frac{Le^{i\Delta \theta }-2he^{i\frac{\beta -\alpha }{2}}}{L-2he^{i\frac{\beta -\alpha }{2}}}\right)$$

Equations (5) and (7) can be used to calculate the elastic strain energy in the following. Figure 5 shows a schematic of a stretching deformation of the beams, excluding the bending and rotation of the beams. The long and short beams are alternately arranged. To consider the long beams as standard, the long beam is fixed at the bottom end. In order for the long and short beams to deform interactively, the top ends of the long and short beams are all connected by stiffening bars. If the entire MPXM is sheared in Figure 4a and the long beam is rotated by Δ*θ*, the short beam is elongated by Δ*l*, as shown in Equation (5). In other words, in Figure 5, the short beam is pulled downward by ∆*l* at the bottom end. In Figure 4a, the long beam is assumed to be rigid, however, in actuality, the long beams are not rigid, and the horizontal rigid bar is balanced at the displacement of *δ* in Figure 5.

Given that the Young’s modulus is *E*, the depth of the beam is *b*, and the thickness of the beam is *t*_beam_, the equilibrium equation for this system is Equation (8).
(8)$$Ebt_{beam}\left(-\frac{\delta }{L}+\frac{∆l-\delta }{l}\right)=0$$

Thus, when the short beam is deformed to be extended by ∆*l*, the rigid block is stable at a position moved down by *δ*. Therefore, the displacement of the rigid bar *δ* can be expressed as a function of ∆*l* in Equation (9).
(9)$$\delta =\frac{L∆l}{L+l}$$
Since the strains of the long and short beams are *δ*/*L* and (∆*l* − *δ*)/*l*, respectively, the change in the elastic strain energy due to the elongation and contraction of the beams per unit cell can be expressed by Equation (10).
(10)$$U_{a}=\frac{Ebt_{beam}}{2}\left\{\frac{\delta^{2}}{L}+\frac{{\left(∆l-\delta \right)}^{2}}{l}\right\}=\frac{Ebt_{beam}{∆l}^{2}}{2\left(L+l\right)}$$
The change in the elastic strain energy due to stretching and contraction is proportional to the square of the overall long beam directional strain.

Since flexure hinges exist at both ends of each beam, the elastic strain energy changes due to the bending of the flexure hinges per unit cell can be expressed by Equation (11)
(11)$$U_{b}=2\int_{0}^{\Delta \theta }M_{ND}d\alpha_{ND}+2\int_{0}^{\Delta \varphi }M_{ND}d\alpha_{ND}\;=\frac{Ebt_{hinge}^{2}}{12}\left[-0.0089+1.3556\sqrt{\frac{t_{hinge}}{2R}}-0.5227{\left(\sqrt{\frac{t_{hinge}}{2R}}\right)}^{2}\right]\left({∆\theta }^{2}+{∆\varphi }^{2}\right)$$
where Equation (1) is used.

The designed MPXM was fabricated with a material extrusion (MEX)-type additive manufacturing (AM) machine (Raise3D Pro2, Raise3D, Irvine, CA, USA) with thermoplastic polyurethane (TPU) with a Young’s modulus of 9.4 MPa. The load to transform the MPXM was applied by hand in this study as a first trial.

## 4. Results

### 4.1. Analytical Results

Figure 6 shows the schematics representing the components of the elastic strain energy per unit cell, which are induced by shear deformation as a function of the rotation angle (Δ*θ*), where *L* = 20 mm, *α* = 120°, *β* = 75°, *b* = 20 mm, *t*_beam_ = 3 mm, *t*_hinge_ = 1 mm, *R* = 1 mm, and *E* = 9.4 MPa. Figure 6a shows the elastic strain energy associated with stretching, i.e., the change in the elastic strain energy due to the expansion and contraction of the beam.

The curve exhibits a w-shaped profile with two equivalent minima. The cell is stable at the inclination angle, showing these two minima. The energy barrier between the two stable phases is large when *h* is large, i.e., when the difference between the length of the long beam (*L*) and the length of the short beam (*l*) is large.

Figure 6b represents the change in the elastic strain energy due to the bending of the flexure hinges. When *h* is large, i.e., when the difference between the length of the long beam *L* and the length of the short beam *l* is large, the deflection angle of the short beam increases more steeply. This effect causes the change in the elastic strain energy due to bending to increase more steeply.

Figure 6c represents the total elastic strain energy change, the sum of the energies due to stretching and bending. When *h* is sufficiently large, the difference in the length between the long and short beams is large, and the energy barrier between the initial phase and the metastable phase is significant, resulting in the emergence of the metastable phase. Conversely, when *h* is small, the energy barrier due to stretching is small, and the barrier in the total energy is overshadowed by the elastic strain energy due to bending, preventing the emergence of the metastable phase by eliminating the minima of the total energy.

As *h* increases, the energy barrier caused by the elastic strain energy due to stretching increases with a higher sensitivity. Thus, the larger the *h* and the greater the difference between the lengths of the long and short beams, the more likely it is that the metastable phase will form. However, there is a limit to this trend because of the constraints to meet the design requirements.

Figure 7 shows the total elastic strain energy (Figure 7a) and its derivative (Figure 7b) with respect to the rotation angle (Δ*θ*) as a function of Δ*θ*. In cases where *h* is 4.5 mm or smaller, there is a local minimum of energy in addition to that for the initial state. The local minimum is represented by the point where the slope d*U*/dΔ*θ* changes from negative to positive with an increasing Δ*θ*. As shown in Figure 7a, when *h* is large, there are two minima: the minimum at Δ*θ* = 0 deg is the initially stable phase, and the minimum at Δ*θ* ≈ 45 deg, where the metastable phase appears after shear deformation. In other words, when Δ*θ* > 0, the derivative is always positive and the MPXM is monostable. Figure 7b shows that the threshold of the MPXM being monostable/bistable is 4.0 < *h* < 4.5.

The derivative of the elastic strain energy shown in Figure 7b corresponds to the shear stress when the MPXM is subjected to a shear strain of Δ*θ*. When the derivative is positive, the MPXM exerts reaction force in the direction that Δ*θ* decreases. When the derivative is negative, the MPXM exerts reaction force in the direction that Δ*θ* increases. Figure 7b suggests that the slopes in the stable and metastable phases are the shear stiffnesses in each phase.

### 4.2. Experimental Results

Figure 8 shows snapshots of the phase transforming of the 3D-printed MPXM. Figure 8a shows the MPXM in the 3D-printed state (i.e., before the phase transformation). Shear strain was applied, and deformation from the initial state to the metastable state was observed row by row, as shown in Figure 8b. When sufficient shear strain was applied, the whole MPXM transformed into the metastable state.

## 5. Discussion

There are several types of martensitic transformation of metallic materials. Some martensitic transformations are classified by stress-induced martensitic transformation, strain-induced martensitic transformation, and deformation-induced martensitic transformation. Strain-induced martensitic transformation (Strn-IMT) has been reported in a paper on the deformation of Co-Cr-Mo alloy. In this section, we discuss the martensitic transformation of the metamaterial we developed, comparing its similarities and differences with actual martensite, as well as exploring the potential developments in the research.

### 5.1. Similarities and Differences with the Classification of Actual Martensitic Transformations

Martensitic transformations can be categorized into several types, as listed below.

(i)Deformation-induced martensitic transformation(ii)Stress-induced martensitic transformation (Strs-IMT)(iii)Strain-induced martensitic transformation (Strn-IMT)

In actual materials, there are transformations such as strain-induced martensite (Strn-IMT) and thermally induced martensite. Sometimes, strain-induced martensite and stress-induced martensite are collectively referred to as processing-induced martensite. It is challenging to distinguish between strain-induced martensitic transformation (Strn-IMT) and stress-induced martensitic transformation (Strs-IMT).

However, since strain formation is manifested by stress, some believe that Strn-IMT should also be classified under Strs-IMT. Stress is the cause and strain and deformation are the effects. Nevertheless, we believe that Strn-IMT should be distinguished from Strs-IMT. Materials manifesting Strn-IMT include alloys like Fe-Mn and Co-Cr-Mo [34,35,36]. For instance, in Co-Cr-Mo alloys, a metastable fcc phase formed by heat treatment does not transform into hcp phase to increase in thickness of hcp phase layer-by-layer (Figure 9) during deformation to a thermally stable hcp phase as the metamaterial in this study did. Instead, a minimum layer of the hcp phase forms from the minimal atomic stacking (i.e., three atomic layers) of the fcc phase at discontinuous and separate locations. These layers then converge to form a thicker hcp region. Within such an hcp phase, stacking faults remain when there is a phase difference in the stacking sequence (i.e., ABAB type stacking on the (0001) plane of the hcp phases). Furthermore, even when the metastable fcc phase remains inside the already-formed hcp phase, the deformation due to the basal plane slip of the hcp phase progresses simultaneously, leading to a minimal increase in the volume fraction of the martensitic phase against the given strain. In other words, a large amount of strain is required to increase the volume fraction of the martensite phase.

### 5.2. Possibility of Thermally Induced Martensitic Transformation

The martensitic transformation of real materials also occurs with temperature. There is martensite generated by a relatively slow temperature change, and thermal martensite introduced by rapid cooling via quenching. Isothermal martensite is generated by holding the temperature at a certain level. TiNi alloys, which are most famous as shape memory alloys, exhibit shape memory characteristics by recovering the shape change due to twinning deformation due to thermally induced martensite transformation. Although the MPXM developed in this study exhibited only stress-induced martensite transformation, it is also possible to develop a metamaterial exhibiting thermoelastic martensite by incorporating a temperature-bending bimetal in the beam. We have already developed and published a TIPXCM that exhibits a permanent compression set through a stress-induced phase transition and recovers the compressive strain by stretching through a thermally induced phase transition [26]. In the TIPXCM study, an Fe-36 mass%Ni Inver alloy (CTE: 3.96 × 10^−6^ K^−1^) with a small coefficient of thermal expansion and Mn-18 mass%Cu-10 mass% with a large coefficient of thermal expansion were used for the curved beam. A clad sheet of %Ni alloy with (CTE: 29.34 × 10^−6^ K^−1^) was used. At high temperatures, the Mn alloy side expanded, and the arc shape with the Invar alloy inside was stable. At low temperatures, the Mn alloy side shrunk, and the arc shape with the Invar alloy outside became stable. In a low-temperature state, both ends of the beam were fixed in the PXCM so that they were point-constrained, and the Mn alloy side was deformed so that the Mn alloy side became the outside of the arc by applying a load, creating a metastable state kept in a compressed state. By raising this temperature, the Mn alloy side of the bimetal expanded, and the curvature of the beam was reversed.

There are several ways of giving a thermal responsiveness to the MPXM developed in this study. If the relative lengths of the beams AC, BD, and P’Q shown in Figure 1 change with temperature, the stable structure can be changed with temperature. If each beam is composed of materials with greatly different coefficients of thermal expansion, and if the difference in the beam length change due to temperature changes is sufficiently large, the stable structure and the metastable structure can be reversed depending on the temperature. However, the MPXM prepared in this study required a large shear force to induce a stress-induced phase transition. In order to create an MPXM capable of recovering its own shape through temperature change, it is necessary to have a mechanism that generates a force corresponding to the shear force by temperature change. For this purpose, it is necessary to incorporate a bimetal and use the force that the bimetal bends due to temperature changes, as used in TIPXCM, to generate a force that exceeds the energy barrier that must be overcome during the phase transition. Structural optimization via a finite element analysis, machine learning, and Generative Design is effective for designing MPXMs, which respond to stress and temperature, such as thermoelastic martensite. As mentioned in the introduction, the authors have already learned the relationship between the structure of PXCM and the load required for a phase transition through a neural network, using a large amount of finite element analysis simulation results, and derived the stress required for a phase transition from the structure. Furthermore, using inverse design, we have succeeded in proposing the design parameters of the lattice that undergoes the phase transition from the stress that causes the stress-induced phase transition using an inverse problem analysis [25].

## 6. Conclusions

In this study, we designed a novel mechanical metamaterial that exhibits martensitic transformation for the future development of metamaterials with the shape memory effect and/or superelasticity. The energy change during the course of the martensitic phase transformation was formulated considering the elastic strain energy associated with the elastic deformation during the course of the phase transformation. The following conclusions were derived.

The metamaterial can transition between two stable configurations via shear deformation. The change in the configuration is similar to that in the atomic arrangement associated with the martensitic transformation of real materials, such as structural changes due to repeated layer-by-layer shear deformation.The energy change governing the transition between the two configurations is tied to the tensile deformation of the short beams, the compressive alteration in the long beams, and the flexing of the connecting hinges. The energy changes in the metamaterial are formulated analytically as a function of the inclination angle associated with the shear deformation.This mechanical metamaterial can be fabricated using MEX-type AM of TPU by utilizing a flexure hinge as an alternative to the linkage connections of the beams. The deformation behavior of the mechanical metamaterial is similar to that of actual martensitic phase-transforming deformation behavior.The outcomes of this study are expected to pave the way for the development of new mechanical metamaterials, such as metamaterials exhibiting thermally induced martensite transformation, the shape memory effect, and superelasticity. This opens the door to potential technological innovations and applications, potentially revolutionizing how we approach design and functionality in various domains.

## Figures and Tables

**Figure 1 materials-16-06854-f001:**
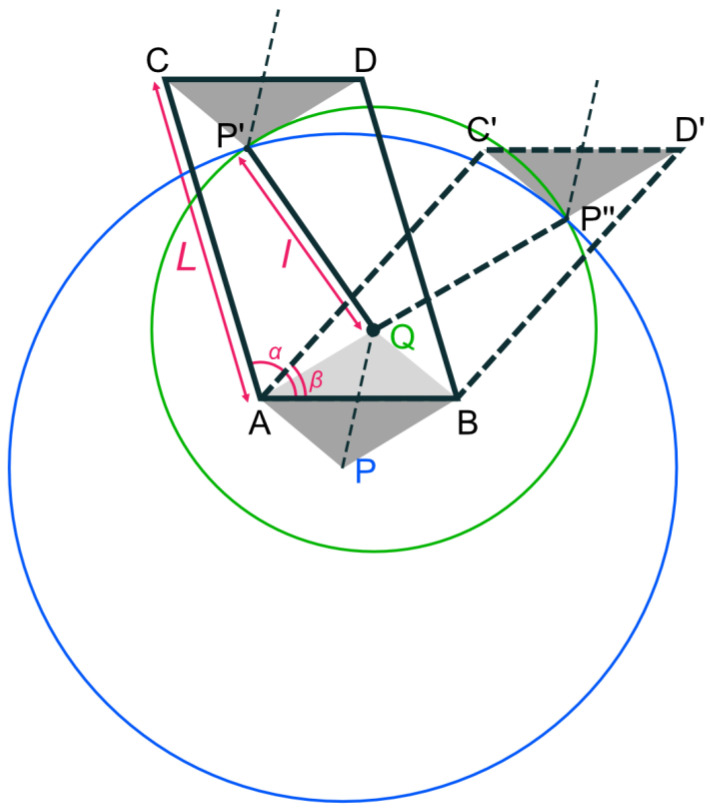
Bistable element of bar-constrained quadrilateral linkage [29]. The bistable element is stabilized in two states: the parallelogram ABDC state and the parallelogram ABD’C’ state.

**Figure 2 materials-16-06854-f002:**
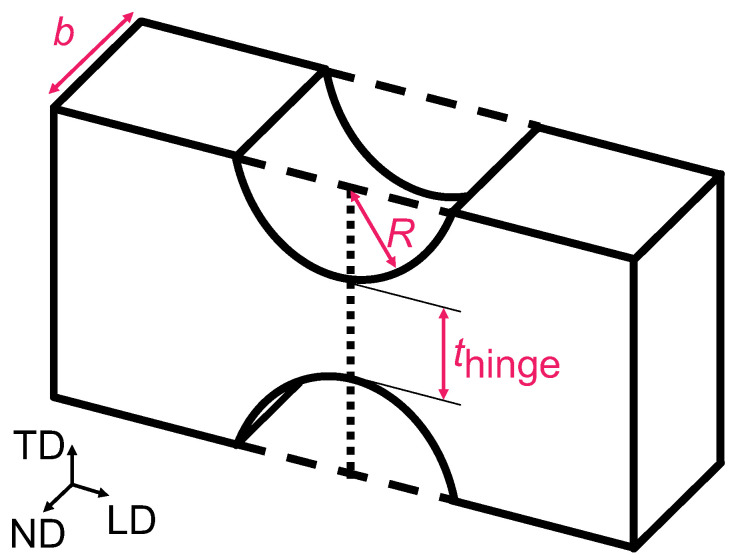
Schematic of flexure hinge. Each side of the beam has circular arcs to make it bend easily. The radius of the arc is *R*, the thickness of the neck of the flexure hinge is *t*_hinge,_ and the width of the beam is *b*.

**Figure 3 materials-16-06854-f003:**
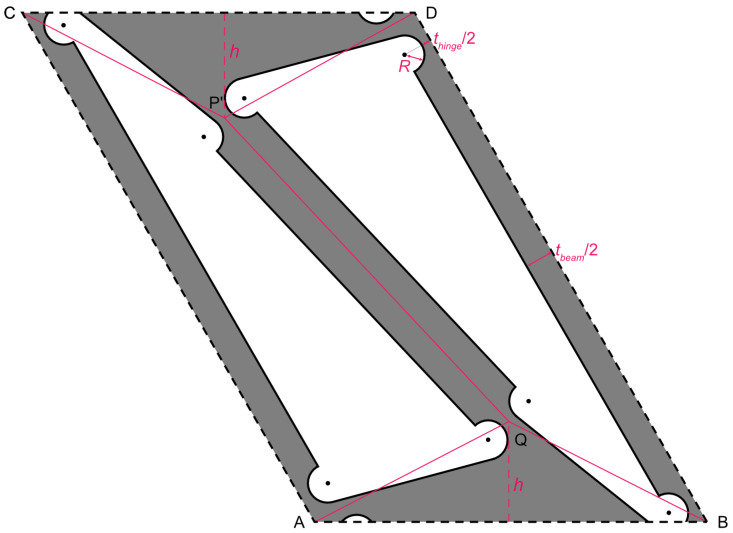
The unit cell of martensitic phase-transforming metamaterial (MPXM).

**Figure 4 materials-16-06854-f004:**
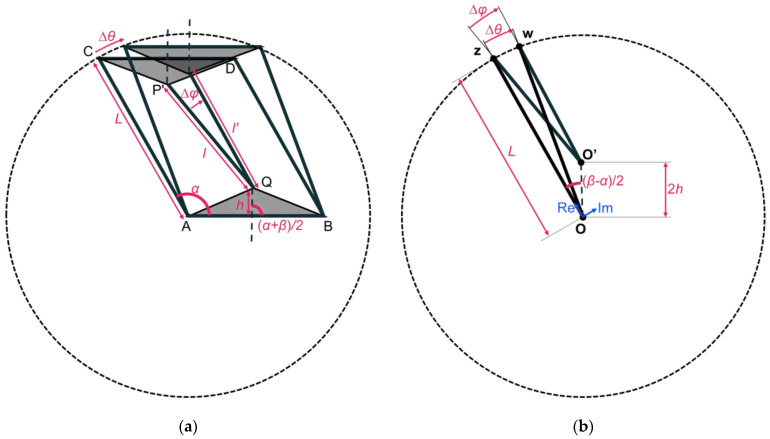
Schematic showing the deformation which occurs accordingly to long beam rotating ∆*θ* around A: (**a**) whole unit cell, and (**b**) principal parts extracted from unit cell.

**Figure 5 materials-16-06854-f005:**
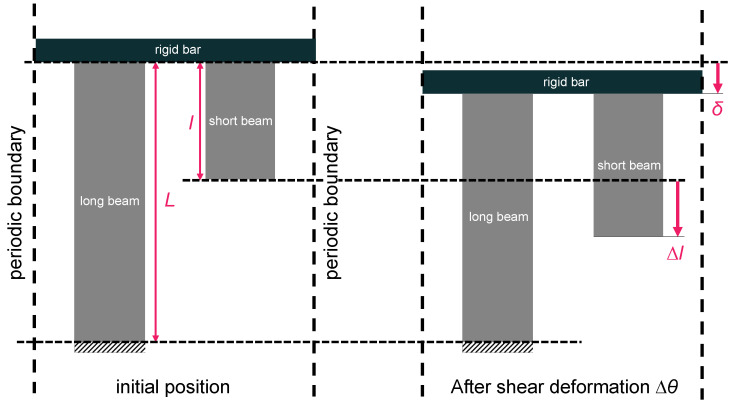
Schematic illustrating the stretching deformation of long beam and short beam, excluding the bending and rotating of beams. Long beams and short beams are aligned alternately under periodic boundary conditions. Long beams are fixed at the bottom ends. Long beams and short beams are all connected with a rigid bar at the top ends. Short beams are pulled downwards by ∆*l* at the bottom ends due to shear deformation associated with the change in the angle *∆θ*. The rigid bar is balanced at a displacement of *δ*.

**Figure 6 materials-16-06854-f006:**
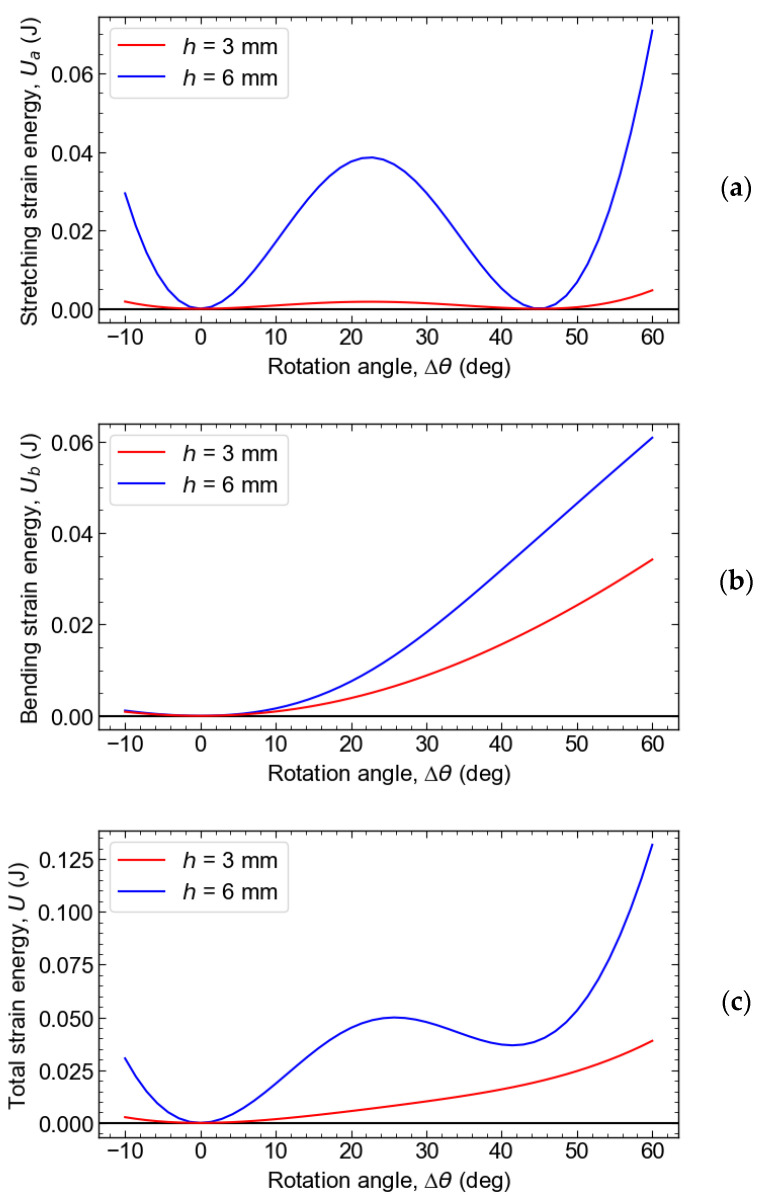
Elastic strain energy per unit cell induced by shear deformation Δ*θ*. (**a**) Elastic strain energy of stretching deformation of the beams, (**b**) elastic strain energy of bending deformation of the flexure hinges, and (**c**) total elastic strain energy.

**Figure 7 materials-16-06854-f007:**
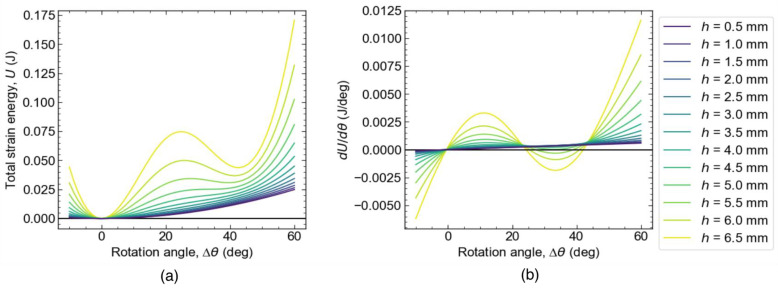
Total elastic strain energy landscapes with respect to rotation angle (Δ*θ*) for various values of *h*. (**a**) Total elastic strain energy plotted against Δ*θ* and (**b**) the slope (d*U*/dΔ*θ*) of total elastic strain energy as a function of Δ*θ*.

**Figure 8 materials-16-06854-f008:**
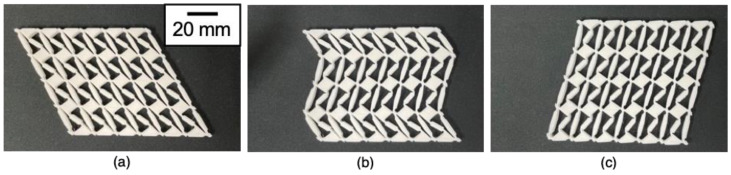
Snapshots of 3D-printed martensitic phase-transforming metamaterial. (**a**) In the initial state, (**b**) the middle two rows transitioned to the metastable state in response to shear deformation, and (**c**) the whole structure transitioned to the metastable state in response to shear deformation.

**Figure 9 materials-16-06854-f009:**
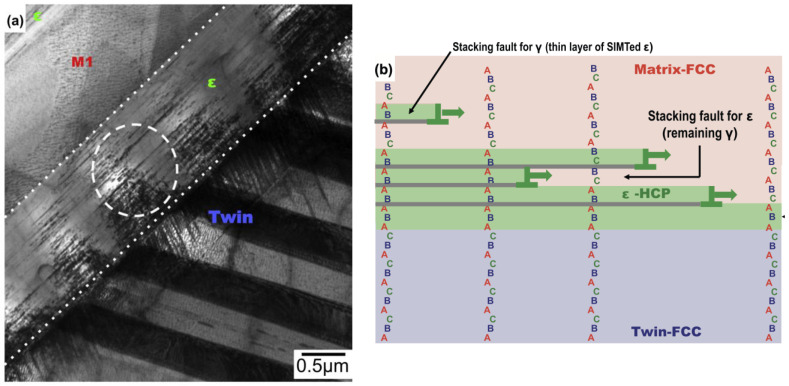
Strain-induced martensite (Strn-IM) of the Co-Cr-Mo alloy [35]. (**a**) Transmission Electron Microscope (TEM) image of ε-hcp phase formed by Strn-IMT of fcc-phase of metastable matrix. (**b**) Schematic illustration showing the change in the stacking of atomic layer on close-packed planes, i.e., {111}-plane of fcc-phase and (0001)-plane of hcp-phase (reproduced with permission from Elsevier.). The dashed circler is indication the location from which the diffraction pattern with streaks indicating the existence of stacking faults were obtained. The characters of “A”, “B”, and “C” represent the location of atoms on {1 1 1} atomic plane. See the reference [35] for more detail.

## Data Availability

Data are available by requesting to the authors.

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
