# Peer review of "Martensitic Phase-Transforming Metamaterial: Concept and Model"

_materials, 2023, doi:10.3390/ma16216854_

Round 1

Reviewer 1 Report

Dear Authors,

The manuscript titled  "Martensitic Phase Transforming Metamaterial" is very interesting.  The article describes a specific group of materials i.e. metamaterials that exhibit deformation behavior characteristic of martensitic transformation, which is the basic prerequisite for specific properties such as the shape memory effect and superelasticity.

The manuscript is clear, relevant for the field and presented in a well-structured manner and it is acceptable for publication in the journal Materials after minor corrections.

Therefore, authors are requested to pay attention to the following:

1. Introduction

 Page 2, Lines from 65 to 67: Please match the font and delete the bracket after "PXCM" in this part:

“…for the development of novel metamaterial. The new PXCM was named multi-axis PXCM). Also, we developed a type of PXCM that can transform in response to the change in temperature by utilizing a bimetal beam that bends spontaneously with increasing temperature [15].”

2. Modeling Approach

 Page 2, Lines 86 and 87: The sizes "L" and "l" which represent, as you state, the length of side AD of the parallelogram, and the length of side AB respectively, do not correspond to Figure 1. Please correct it.

 Page 2, Line 92: Correct the marking of the triangle OAB according to Fig. 1. which you refer.

 Page 3, Line 109: Please state the meaning of symbols "alpha", "M" and "E" which occur in Equation (1).

 Page 4, Line 117: Please delete unnecessary text ("The red line in the fig.").

 Page 4, Lines 118 and 119: Correct the midpoint symbol "O" and the length "OP" in accordance with Fig. 3.

 Page 5, Line 128: Please make sure that the point O' in this sentence is consistent with the Fig. 4a to which you refer.

 Page 7, Line 180: State the meaning of the symbol "d" in Equation (11).

 Page 7, Lines from 182 to 187: An almost identical description of Fig. 5 is given at page 6. Please rephrase the text.

3. Results

 Page 10, Lines 241 and 242: Please correct the first sentence in the chapter 3.2 Experimental results.

("Figure 8 shows snapshots of 3D-printed MMPXM MMPXMwas 3D-printed as shown in Figure 8a.") ?

 Page 10, Line 244: Correct the typo (“…wholMPXMXM”…).

4. Discussion

 Pages 10 and 11: Please check the accuracy listing of the abbreviations for stress induced martensitic transformation and strain induced martensitic transformation (Strn-IMT and Strs-IMT) in chapter 4.1.

 Pages 11 and 12: When citing the word "Martensitic" in chapter 4.2, the uppercase letter should be replaced by a lowercase letter in several places.

5. Conclusions

 Pages 12 and 13, Line 339 and 356: see previous comment

 Pages 13, Line 353: Please explain of the abbreviation “AM” in that sentence.

Refrerences

Please update the literature with more recent references. A grate number of the mentioned references are older than 5 and even 10 years.

Missing page numbers (first and last page) for references [2] and [15]. Please complete.

Please check if it is possible to cite Graduate Theses and Dissertations (and several different ones at that) as a reference [16] (only one reference?).

 Reviewer

Reviewer 2 Report

Dear Authors

The article presented for review is very interesting. A very good concept of the material was presented, a good theoretical study and a valuable discussion. However, the article contains some weak elements that I think should be improved. Therefore, I am posting a few comments below: 

1. The title of the article suggests that new material has been developed. In fact, a material concept and model were developed. Therefore, I suggest that the word "model" be included in the title of the article. 

2. Chapter 3.2 - The experiment was described in too little detail. It is necessary to describe in more detail the preparation of samples and the execution of the experiment. If 3D printing was used, it is important in what orientation the samples were made. What factor caused the deformation? 

3. Line 270. My view is as follows (Authors do not have to agree with it) - deformation is an effect, and stress is the cause that causes deformation. Stress in the material may come from external forces or may be caused by other factors, e.g. temperature. Then thermal stresses arise and result in deformations. 

Kind Regards

Reviewer

Round 2

Reviewer 2 Report

Dear Authors

Thank you for the corrections made according to my suggestions.

Kind Regards

Reviewer